# Direct Observation of Feeding Behavior of Adult Tabanidae (Diptera) on Beef Cattle from Khon Kaen Province in Thailand

**DOI:** 10.3390/insects15080602

**Published:** 2024-08-10

**Authors:** Kanchana Thinnabut, Wanchai Maleewong, Ubon Tangkawanit

**Affiliations:** 1Department of Entomology and Plant Pathology, Faculty of Agriculture, Khon Kaen University, Khon Kaen 40002, Thailand; kanjana_ti@kkumail.com; 2Department of Parasitology, Faculty of Medicine, Khon Kaen University, Khon Kaen 40002, Thailand; wanch_ma@kku.ac.th; 3Mekong Health Science Research Institute, Khon Kaen University, Khon Kaen 40002, Thailand

**Keywords:** tabanid, behaviors, feeding

## Abstract

**Simple Summary:**

An understanding of tabanid alighting and feeding behavior on vertebrate hosts may result in more effective disease management and control strategies. The purpose of this study was to determine the tabanid behavior of beef cattle. The alighting behavior, time of day, duration of tabanid feeding on beef cattle, as well as variables associated with tabanid infestation, were recorded. According to the results, the lower leg was the most tabanids infested location. The greatest number of tabanids landing on cattle occurred at midday. Temperature was the most important factor for tabanids landing on hosts. To forecast the tabanids’ reactions to environmental factors, regression analysis was carried out.

**Abstract:**

Tabanidae (horse flies and deer flies) are hematophagous insects that cause direct and indirect damage to animal production. The aims of this study were to determine the preferred site, time of day, and duration of tabanid feeding on beef cattle and identify factors related to infestation by tabanids. The population of tabanids was surveyed on certain body parts of the beef cattle (fore udder, tail, navel, leg, dewlap, body, and under) during the morning hours (9.00–10.30 a.m.), midday (12.00–13.30 a.m.), and afternoon (15.30–17.00 p.m.) every day for 10 days. The findings showed that two genera, *Tabanus* Linnaeus, 1758, and *Chrysops* Meigen, 1803, landed on the cows. The leg was statistically significantly the most frequent landing site for tabanids (15.067 ± 7.54) compared with other parts. The average feeding duration for each insect was 2.76 ± 1.77 min. The results showed that a significant number of tabanids were present during midday, as compared with the morning and afternoon. Temperature was found to be positively associated with fly abundance. A regression model was derived in this study (y = 4.23x − 116.09). This information is important for tabanid control and prevention in beef cattle.

## 1. Introduction

Tabanids, sometimes known as horse flies and deer flies, belong to the Tabanidae family in the order Diptera [1,2]. Over 4400 species have been described throughout the world, classified into the subfamilies Chrysopsinae, Pangoniinae, Scepsidinae, and Tabaninae [1,3,4,5,6]. In Thailand, three families (Tabaninae, Chrysopsinae, and Phlebotominae) and four genera (*Tabanus* Linnaeus, 1758, *Atylotus* Osten Sacken, 1876, *Chrysops* Meigen, 1803, and *Haematopota* Meigen, 1803) have been reported [7,8,9]. *Tabanus megalops* was the most abundant species in beef cattle [10].

Tabanids are annoying pests that directly affect both humans and animals because of their incessant, painful biting and loud flight. The large blood meal size (20–600 μL/blood meal) ingested by tabanids is associated with notable blood losses [1]. In addition to these direct consequences, tabanids indirectly cause significant costs by transmitting infections from one host to another. They can function as mechanical vectors, for example, *Anaplasma marginale*, *Trypanosoma evansi*, and lumpy skin disease virus [10,11,12], and as biological vectors of pathogenic parasites such as *Loa loa* and *Trypanosoma theileri* [13,14].

One of the most significant economic animals driving Thailand’s economic expansion is beef cattle. In line with the rising popularity of beef consumption, it is one of the biggest agricultural sectors and a major source of revenue for both farmers and the nation. In 2023, the national herd reportedly contained 69,011 cattle with an estimated value of USD 41.03 million [15]. The production of beef cattle in Thailand has decreased by as much as 1.24% in 2020 and 2021 because of the emergence of a new lumpy skin condition. Produce was severely damaged by this disease, and a large portion could not be sold or transported to the slaughterhouse [16]. The carriers of important vector-borne diseases that affect cattle (e.g., mosquitoes, tabanids, and stable flies) are widespread around the globe and involve a large variety of higher taxa and species [17]. Tabanids, one of the most significant vectors, are hematophagous insects when compared to many other dipterans. The broader proportions of the mouth proboscis enable them to take in a greater amount of blood. Additionally, as carriers of bacteria, viruses, and parasites, these insects are important for medicine and veterinary care [18,19].

To promote the growth of the beef cattle industry, it is crucial to prevent disease in cattle and lower the number of disease carriers. Understanding the biology, behavior, preferred feeding site, and feeding time behavior of tabanids will provide an understanding essential to the application of preventative and elimination measures. There have been some previous investigations of the behavior of tabanids on cattle. For example, Muzari et al. [20] revealed that fly attack rates were influenced by host species, and Brady and Shereni [21] reported that the color of the host had an impact: darker targets were observed to experience greater damage than lighter ones [21]. Baldacchino et al. [22] found that different species of tabanid landed preferentially on the different parts of the cow’s body. Herczeg et al. [23] reported that weather variables influenced the flight activity of horseflies. Additionally, Horváth et al. [24] found that horseflies are attracted to targets that are uniformly black; sunlit, warm, shiny, and dark targets are preferred by *Tabanus tergestinus*.

In this study, the preferred site, time of day and duration of tabanid on cattle were investigated. Feeding duration affect the quantity of blood that insects consume [22] as well as the potential for disease transmission. The preferred feeding site on the host of the insect vector is important for insect surveys and alerts. According to Muzari et al. [20], a deeper comprehension of tabanids’ alighting and feeding habits on vertebrate hosts could result in enhanced disease management and intervention tactics. The biting behavior of tabanids on cattle is poorly known in Thailand, even though there have been previous studies in other countries. The situation in Thailand involves different tabanid species, geographical locations, and weather. The purpose of this study was to examine the feeding time period and preferred site of tabanids landing on beef cattle related to environmental variables in order to provide data useful for the regulation, prevention, and control of insect vectors in the Thai beef cattle sector.

## 2. Materials and Methods

### 2.1. Study Site and Cattle

The experiment was conducted in upper-northeastern Thailand, a location where traditional beef-cattle farms are especially widespread [25]. The study site was located in village No. 9, Ban Non Muang sub-district of Sila in Muang Khon Kaen District, Khon Kaen province. There are five villages in Ban Non Muang (village No. 5–9), with thirteen herds of cattle. The selected farm was located at 16°28′58.08″ N 102°48′13.78″ E. (elevation 210 m). There were five cattle (local Esan breeds) on this farm with a 1–4-year age range. The local geography of the herd location comprises an open area with meadow and many houses scattered in the area. There are many rows of banana trees on the east of the farm, a big tree (Samanea saman) in the center, and several tiny ponds (1–3 m in diameter) in the farm area. No pesticides are used on this farm. A preliminary study had found many tabanids at this site.

### 2.2. Tabanid Landings

A total of three mature female cattle (age range of 2–4 years) were examined. The experiment employed female beef cattle due to the aggressive nature of males and the lower ratio of males to females in Thai farms (about 1:4 in Khon Kaen Province) [26]. Tabanid landings and host defensive behavior were observed three times a day: morning (09:00–10:30 a.m.), midday (12:00–13:30 a.m.), and afternoon (15:30–17:00 p.m.). The study was conducted on sunny days. Three cattle were monitored for each time period (morning (3 cows), midday (3 cows), and afternoon (3 cows)); for each period, 1 cow was allocated for monitoring. The number of tabanid species landing on cattle during a 30 min period was recorded from the body parts of cattle (fore udder, tail, navel, leg, dewlap, body, and under) (Figure 1). In the experiment, one observer observed the cows from one side. The observer remained continuously near the cattle (1–3 m) using their unaided eyes and sometimes employing a magnifying lens on the mobile phone. Feeding duration of tabanids on the cattle was recorded. Hematophagous insects bite as soon as the skin is punctured by their mouthparts. However, occasionally the cattle’s repulsive behavior causes them to be interrupted and prevent them from feeding. Consequently, feeding duration started to record at the moment when the abdomen contracted and expanded, indicating that blood was pumping to the body. The temperature and relative humidity were measured using a digital thermo-hygrometer (TL8039/JEDTO). The study was conducted using 10 replications (on 10 days) during October–November 2023.

Some specimens of tabanids were carefully captured. Cattle were scared when a sweep net was employed; some of them became hostile. After being observed for half an hour, insects were collected from the cattle. The flies were captured using a transparent plastic bag that had dimensions of 13 × 20 cm. A plastic bag encased the tabanid that was sucking on the cattle’s body. When it flew up, the insect was captured with a plastic bag and identified to its genus using the key of Burton et al. [8].

### 2.3. Statistical Analysis

The data of tabanids alighting at different sites on the cattle were analyzed using a completely randomized design (CRD) that was employed for one-way analysis of variance. Significance levels were set at 5%, computed in Statistics X [27]. The number of tabanids landing on the cow’s legs (upper, lower, fore, and hind) per day was analyzed using two sample *t*-test at the confidence level of 95% computed in Statistics X [27].

Environmental data for temperature (X1) and relative humidity (X2) were recorded as independent variables. Analysis of variance (ANOVA) was conducted for tests of significance if at least one of weather variables was linearly related to the response variable. The null hypothesis states that *β*_1_ = *β*_2_ = … *β*_i_ = 0. If the alternative hypothesis is accepted (*β* ≠ 0), then a mathematical model of tabanids can be analyzed by logistic regression. The model can be described as:y = *β*_0_ + *β*_1_*X*_1_ + *β*_2_*X*_2_ + … + *ε*(1)
when y is number of tabanids. *β*_0_, *β*_1_, *β*_2_ … *β*_i_ are coefficients, *X*_1_, *X*_2_, *X*_3_ … *X*i are independent variables, and ε is a random error component.

The model was validated quantitatively by the calculation of the correlation coefficient between the predicted data and observed data (r^2^) and root mean square error (RMSE).

## 3. Results

### 3.1. Landing Sites of Tabanids on the Cattle

Tabanids found landing on cattle belonged to two tribes: Tabanini (genus Tabanus) and Chrysopini (genus Chrysops). A total of 589 females of Tabanus (99.49%) and 3 females of Chrysops dispar (0.05%) were detected during the 10 days of observations. There were four species found in the study (Tabanus megalops, T. rubidus, T. striatus, and T. eurytopus). However, they are different from each other by the different patterns and colors in the abdomen, which were difficult to distinguish during the experiment. With uncertainty, the results do not include the species name in the results. The average number of tabanids alighting on cattle was 6.54 per 30 min (0.22 per min). The results showed that legs were the most frequent landing sites (76.35%) followed by the body (14.36%), under (6.25%), tail (0.84%), and navel (0.68%) (Table 1). The tabanid landing rate on the lower part of the legs was significantly different from the upper (84.29% and 15.70%, respectively); however, rates for the forelegs were not significantly different from the hind legs (Table 2).

### 3.2. Time Period and Meteorological Factors of Tabanids Landing on Host

The results indicate that tabanids landed on cattle during all three observation periods: in the morning (09.00–10.30), at midday (12.00–13.30), and in the afternoon (15.30–17.00). Tabanids started to land on the cattle around 10 a.m. The highest number landing on cattle was recorded at midday (11.87 ± 5.88). The difference between midday and the morning and afternoon figures was statistically significant (*p* < 0.01) (Table 3).

Temperature and relative humidity data are presented in Table 4. The results indicated that meteorological factors play a significant role in the number of tabanids landing on hosts. The regression equation of the tabanid population with measured parameters related to temperature (°C) and relative humidity was analyzed. Statistical analyses (ANOVA) indicated that β ≠ 0, which does not support the null hypothesis (Table 5). Therefore, at least one variable was related to the populations of tabanid species. The regression model revealed significant association only with temperature (*p* < 0.05) (Table 5, which was found to be positively associated with fly abundance. The regression model was as follows:y = 4.23x − 116.09(2)
x = Temperature (°C)

The model had r^2^ = 0.763.

Additionally, it was observed that certain tabanids fed on less cattle on windy days than they did on other days.

### 3.3. Feeding Duration

During 10 days of observation on three cattle, 53 tabanids were found to spend time with the host. The results showed that the longest duration of tabanid feeding was 10.25 min, while the shortest was 1.00 min (average 2.76 ± 1.77 min/insect). The frequency of feeding duration is shown in Figure 2. These findings included tabanids ceasing to suck blood because they were fully engorged and those that were interrupted by the host. The host would respond to tabanids biting by licking their backs, twitching their skin, flicking their tails, or stamping their legs, which resulted in interruption.

## 4. Discussion

### 4.1. Tabanid Landing Sites on the Cattle

The results indicated species of Tabanini as key pests of cattle on the studied farm; thereby corroborating the reports of Baldacchino et al. [22] and Changbunjong et al. [28]. The high proportion of tabanids landing on the cattle’s legs was consistent with the findings of Baldacchino et al. [22] and Raymond and Rousseau [29] and similar to the reports of Muzari et al. [20] regarding the percentage of tabanids landing on the legs of horses (70–100%). A previous study indicated that the small *Tabanus* species frequently land on the back, while the larger species visit the legs, flanks, or lower body [30]. However, the results of this investigation indicated that large species landed on the leg.

The number of Tabanini landing on the hind legs was not significantly different from those on the fore legs. However, Baldacchino et al. [22] reported that Tabanini landed preferentially on the forelegs (55.8%), rather than the hind legs (19.2%), while Lendzele et al. [31] reported that *Tabanus* were most frequent on hind legs. Some reports have indicated that tabanids prefer to land on the lower sections of the body (belly and legs) of animals, such as horses and pigs, and landed upon the rear sections (hind legs and tail) of kangaroos [20]. According to our observations, some tabanids landed on body sites of cattle when they were lying down (41.18% of 85 tabanids). It is our assumption that many tabanids favor activity close to the ground. Krcmar and Maric [32] found that the lower legs of a humans were the preferred feeding sites for *Tabanus* spp.

The behavior of tabanids landing on the lower leg is consistent with the findings of Altunsoy [33], who stated that *Tabanus* species favored their hosts’ foot. Numerous factors have been suggested as potentially influencing feeding location: (1) host-repelling behavior; (2) physiological aspects of the body part, and (3) decreased inter-specific competition.

The number of tabanids landing on the lower part of the legs may be related to the cow’s ability to repel insects. Since tail switches can only protect the upper portion of the hind legs and head throws can protect the upper leg rather than the lower legs of the foreleg, this may explain why tabanids tend to land on lower legs more often than upper legs.

For physiological aspects, Todd [34] suggested that some insects may find it better to bite legs due to the thinner skin and closer proximity of blood capillaries to the skin’s surface. Additionally, Mullens and Gerhadt [35] suggested that differences in hair length on different parts of the cattle’s body relative to the length of tabanid mouthparts might be important. Previous studies have demonstrated that flies prefer to land upon parts of cattle, such as the head, which expose areas of CO_2_ emissions [35]. However, considering the tabanids landing on the leg reported here, the hypothesis must be discounted.

According to Waage and Davies [36], adult bloodsucking insects share host blood as a resource and frequently display niche partitioning that may indicate interspecific competition. In a previous study, it was found that, 90% of tsetse flies (*Glossina* spp.) landed on the belly, forelegs, and hind legs, while *T. par* Walker often landed on the back [37]. Lendzele et al. [31] reported *Stomoxys* predominant on the legs and belly region, whereas *Simulium* preferred landing around the under/scrotum and cow breast.

The results from this research showed that *C. dispar* species landed on the navel, body, and upper leg, corresponding to the report by Lendzele et al. [31]. However, this result is different from the report of Miletti et al. [38], which indicated *Chrysops* landing on the heads of horses. Therefore, the landing site of *Chrysops* may vary depending on the host species. Mullens and Gerhadt [35] suggested that *Chrysops* tended to attack areas of thin hair and easily accessible skin.

### 4.2. Time of Day and Meteorological Factors of Tabanids Landing on Host

There was a significant difference in the number of tabanids that landed on cattle at midday compared to the morning and afternoon. These results are different from the study of Baldacchino et al. [22], who reported that there was no significant difference for Tabanini in the morning (9.00–10.30), midday (12.00–13.30), and afternoon (15.30–17.00). However, our results correspond to those for Haematopini in the same report, which was most prevalent at midday.

Temperature was positively correlated with both flight activity and tabanid abundance, which is similar to the findings of Herczeg et al. [23] and Baldacchino et al. [23]. Increased temperatures were mainly related to an increase in tabanid landings. Amano [39] noted that cold temperatures tend to limit the initiation of flight activity. Horváth et al. [24] suggested that, when tabanids are in a cool area (a shady, dark, or bright host (because brighter surfaces absorb less sunlight than darker ones do)), the cooling process of wing muscles is active, so a bloodsucking tabanid cannot immediately escape when the host tries to swat them. In comparison, in a heated area (in sunlight or a dark host), the cooling process of wing muscles is inactive, so when the host tries to swat a bloodsucking female tabanid, it can take flight quickly. These results are consistent with the observations that tabanids like to feed on a sunny day and on dark hosts.

From the results, fly abundance was not related to the relative humidity. According to Herczeg et al. [23], tabanids were indirectly affected by the relative humidity of air through air temperature. The trapping of tabanids was optimal at a relative humidity of 35% (temperature of about 32 °C). At a relative humidity of 80% (temperature less than 18 °C), tabanids could not be trapped. In the present study, such environmental conditions did not occur.

Light intensity is a factor related to the biting behavior of insects. McElligott and Galloway [40] suggest that low light intensity levels are associated with low flying activity in horse flies. The light intensity changed throughout the day; it increased during the morning, peaked at noon, and then started to decline in the afternoon, which correlates with tabanid activity in the results.

Additionally, it has been observed that tabanids less actively feed on cattle on windy days than they do on other days. Gibson and Torr [41] found that high wind speeds make it more difficult for insects to fly, especially when the wind speed was greater than the insect’s maximum speed. Wind also influences an insect’s capacity to detect olfactory cues in the surrounding environment.

### 4.3. Feeding Duration

Based on the findings, tabanids often spend approximately two minutes with the cattle. Chvala et al. [42] reported that Tabanids can consume up to 200 mg of blood in a few minutes. Some of them stopped feeding before they reached complete engorgement due to cattle’s repellent behavior, which also corresponds with the results of a study of Caro [43] on zebras. From our observations, there was an initial behavioral response of the cattle; however, the cattle showed no defensive response if the tabanids were able to remain on the body for more than 10–15 s. The anesthetic substances in the saliva of tabanids could be the reason of this. Strong vasodilatory and anticoagulant properties are known from tabanid saliva, along with a wide range of inhibitors of platelet aggregation [44]. Blood remains flowing from the wound after the tabanids moved away from the cattle. This increases the possibility of a visit by another fly, thereby increasing the risk of blood loss and infection with pathogens.

## 5. Conclusions

In conclusion, the lower part of the hind leg was the most infested area for *Tabanus*. The greatest number of tabanids landing on cattle occurred at midday. Temperature was a meteorological factor that played a significant role in tabanids landing on hosts. Regression analysis was performed in order to predict responses of tabanids to environmental variables. This information is helpful for tabanid prevention and control. However, the further study of additional environmental factors is needed.

## Figures and Tables

**Figure 1 insects-15-00602-f001:**
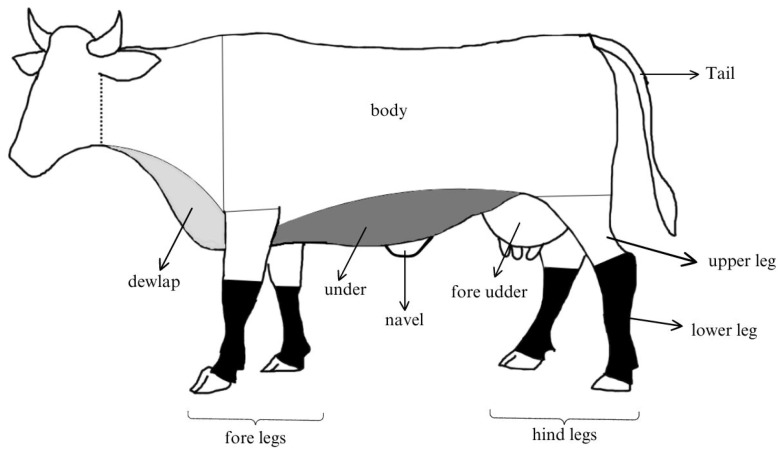
The body parts of cattle where observations were focused: fore udder, tail, navel, leg (upper and lower), dewlap, body, and under.

**Figure 2 insects-15-00602-f002:**
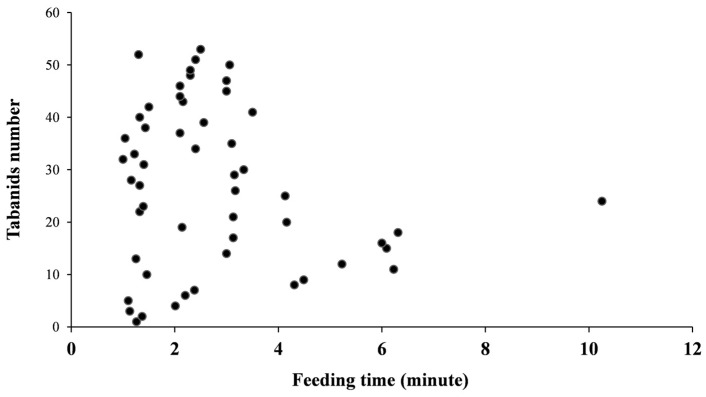
The frequency of feeding duration of tabanids on cattle.

**Table 1 insects-15-00602-t001:** Number of tabanids landing on different parts of cattle per day.

Body Part	Number of Tabanid Landing (Mean ± SD)	%
Fore udder	0.033 ± 0.11 ^c^	0.17
Tail	0.167 ± 0.24 ^bc^	0.84
Navel	0.133 ± 0.23 ^bc^	0.68
Leg	15.067 ± 7.54 ^a^	76.35
Dewlap	0.267 ± 0.21 ^bc^	1.35
Body	2.83 ± 2.86 ^b^	14.36
Under	1.23 ± 1.30 ^bc^	6.25
Total	19.727 ± 11.81	100
*p* (0.05)	*p* < 0.001	
CV (%)	109.64	

Different letter(s) after the mean values indicate significant difference among each treatment tested by LSD (*p* < 0.05).

**Table 2 insects-15-00602-t002:** Number of tabanids landing on leg (upper, lower, fore, and hind) of the cattle per day.

Leg	Tabanid (Mean ± SD)	%
Upper	2.37 ± 1.76	15.70
Lower	12.70 ± 6.22	84.29
	*p* = 0.0004 *	
Fore	19.7 ± 9.29	43.58
Hind	25.4 ± 13.49	56.41
	*p* = 0.1356 ^NS^	

(NS) is non-significant and (*) indicates significant difference at *p* < 0.05 by *t*-test (*p* < 0.05).

**Table 3 insects-15-00602-t003:** Time span and number of tabanids landing on cattle.

Time Span	Number of Tabanids Landing on Cattle (Mean ± SD)
9.00–10.30	5.03 ± 3.19 ^b^
12.00–13.30	11.93 ± 5.88 ^a^
15.30–17.00	2.77 ± 1.80 ^b^
*p* value	0.0001 *
CV (%)	61.30

(*) indicates significant difference at *p* < 0.05. Different letter(s) after the mean values indicates significant difference among each treatment tested by LSD (*p* < 0.05).

**Table 4 insects-15-00602-t004:** Temperature and relative humidity during the study.

	Temperature (Mean ± SD)	Relative Humidity (Mean ± SD)
Morning (09.00–10.30)	29.22 ± 2.65	55.9 ± 12.75
Midday (12.00–13.30)	31.26 ± 1.92	46.2 ± 17.13
Afternoon (16.30–17.00)	29.88 ± 2.10	50.3 ± 15.96

**Table 5 insects-15-00602-t005:** ANOVA F-test statistics for the experiment studying influence of weather variables on tabanids.

	df	SS	MS	F	Significance F
Regression	2	710.08	355.04	11.26	0.01 *
Residual	7	220.76	31.54		
Total	9	930.83			

(*) indicates significant difference at *p* < 0.05.

## Data Availability

Dataset available on request from the authors.

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
