# Peer review of "Direct Observation of Feeding Behavior of Adult Tabanidae (Diptera) on Beef Cattle from Khon Kaen Province in Thailand"

_insects, 2024, doi:10.3390/insects15080602_

Round 1
Reviewer 1 Report
Comments and Suggestions for Authors
Title: The phrase "feeding time" in this context is ambiguous and may be misunderstood as "feeding duration." Please consider finding more suitable wording.
Material and Methods:
Line 102: Please elaborate on why the experiment was conducted only on mature female cattle. Were the males too aggressive?
Lines 106-114: The behavior observation section is unclear. Please provide additional details.
How many observers were used in the experiment? If the authors observed small flies on a cow from one side, please specify this in the methods.
While observing landing sites with the unaided eye is understandable, it seems challenging to accurately observe biting behavior, especially in difficult areas such as underneath the cattle.
Line 113: "The feeding duration of tabanids on the cattle was recorded." How did you determine that the flies were exhibiting biting behavior? How did you ensure they began biting to record the feeding duration? When did you start recording the biting period?
Line 114: Why did you use a plastic bag to capture the flies instead of a sweep net? Was it to prevent other flies on the cow's body from escaping? When did the collection occur? After 30 minutes of observation? Please provide more details about the fly collection process as it might be useful for other researchers.
Fig. 1: Is it possible to shade the areas under observation?
Results:
Lines 139-141: Italicize the genus names.
Lines 178-185: Is there a difference in feeding times at different feeding sites?
Fig. 2: What do "Lower" and "Higher" refer to? Please provide more definition.
Moderate editing of English language required
Author Response
Comments 1: Title: The phrase "feeding time" in this context is ambiguous and may be misunderstood as "feeding duration." Please consider finding more suitable wording.
Response 1: The title has been changed to “Direct observation of feeding behavior of adult Tabanidae (Diptera) on beef cattle from Khon Kaen Province in Thailand”. Feeding duration refers to the amount of time that tabanids spend on the host, while feeding time refers to the time of day. Wording has been rechecked the meaning and corrected in the text
Comments 2: Material and Methods Line 102: Please elaborate on why the experiment was conducted only on mature female cattle. Were the males too aggressive?
Response 2: The experiment employed female beef cattle due to the aggressive nature of males and the lower ratio of males to females in Thai farms (about 1:4 in Khon Kaen Province) (Department of Livestock Development, 2023). (Line 103-105)
Comments 3: Lines 106-114: The behavior observation section is unclear. Please provide additional details.How many observers were used in the experiment? If the authors observed small flies on a cow from one side, please specify this in the methods.
Response 3: The cows in the experiment were observed only from one side by the single observer. The materials and procedures now included more details. (line 112)
Comments 4: While observing landing sites with the unaided eye is understandable, it seems challenging to accurately observe biting behavior, especially in difficult areas such as underneath the cattle.
Response 4: For the observation of tabanids, a distance of 1–3 meters is enough for the naked eye. A mobile phone camera with a magnifying lens was used for the detection of insects on cattle. The information was added to the material and methods, as shown in line 112-114. In case, it was landed underneath the cattle. It was detected by the observer.
Comments 5: Line 113: "The feeding duration of tabanids on the cattle was recorded." How did you determine that the flies were exhibiting biting behavior? How did you ensure they began biting to record the feeding duration? When did you start recording the biting period?
Response 5: Hematophagous insects bite as soon as the skin is punctured by their mouthparts. However, occasionally the cattle's repulsive behavior causes them to be interrupted and prevent them from feeding. Consequently, feeding time started to record at the moment when the abdomen contracted and expanded, indicating that blood was pumping to the body. (Line 114-119)
Comments 6: Line 114: Why did you use a plastic bag to capture the flies instead of a sweep net? Was it to prevent other flies on the cow's body from escaping? When did the collection occur? After 30 minutes of observation? Please provide more details about the fly collection process as it might be useful for other researchers.
Response 6: Cattle were scared when a sweep net was employed; some of them became hostile. After being observed for half an hour, insects were collected from the cattle. The flies were captured using a transparent plastic bag that had dimensions of 13 ´ 20 cm. A plastic bag encased the tabanid that was sucking on the cattle's body. When it flies up, insects will be captured. (line122-126)
Comments 7: Fig. 1: Is it possible to shade the areas under observation?
Response 7: Figure 1 is modified, the areas are shaded.
Comments 8: Lines 139-141: Italicize the genus names.
Response 8: The genus name was italicized. (Line 161-163)
Comments 9: Lines 178-185: Is there a difference in feeding times at different feeding sites?
Response 9: I think reviewer means feeding duration at different feeding site. Unfortunately, we do not have enough information to find significant differences among the sites.
Comments 10: Fig. 2: What do "Lower" and "Higher" refer to? Please provide more definition.
Response 10: The words “lower” and “higher” in the Fig. 2 was deleted.
Reviewer 2 Report
Comments and Suggestions for Authors
The Manuscript is well written and a thorough literature search has been conducted.
In my opinion, the following points could be improved.
The first would be the question of the study design. Observing three cows over 10 days seems to me to be a very small number, which may influence the results. It would have been better to observe the tabanids over a longer period of time in order to gain more information, given the seasonality of these species.
Another point is the identification of species at genus level only. Here it would have been potentially very informative to diversify the few species that occur on site using cameras or even flight sounds in order to obtain more precise results at species level. For example, perhaps one species only lands on the body of cattle, while others prefer the legs.
A further point would have been to record the light intensity in the area under investigation, as sky coverage influences the activity of the Tabanids.
Another very interesting point would be a comparison of differently colored cattle and the effect on the attacks of tabanids.
All in all, this manuscript gives enough results to show how tabanids can be effectively kept away from cattle or how their nuisance can be reduced with a minimum of resources.
Author Response
Comments 1: The first would be the question of the study design. Observing three cows over 10 days seems to me to be a very small number, which may influence the results. It would have been better to observe the tabanids over a longer period of time in order to gain more information, given the seasonality of these species.
Response 1: We designed the experiment with the purpose of examining the feeding time, feeding position, and feeding duration of certain common tabanid species. Diversity and seasonality are therefore excluded from the investigation.
The cow's landing and feeding behaviors were extensively monitored during the trial. Consequently, in order to reduce bias, just one individual examined the data. For each time period with ten replications (10 days), three cows (samples) are sufficient for close observation. Using more animals will extend the time period, which could result in significant variations in the temperature or any other abiotic factor during that time.
Sample Size in the Study of Behavior by Taborsky (2010) states that a Type II error, also known as a "false-negative result," is more likely to occur when there is a small sample size. The variance in the data is reduced by prudent sampling and maximizing accuracy in parameter estimation, e.g., by using repeated measures instead of randomized groups. This experiment we repeated for 10 replications (10 days).
Comments 2: Another point is the identification of species at genus level only. Here it would have been potentially very informative to diversify the few species that occur on site using cameras or even flight sounds in order to obtain more precise results at species level. For example, perhaps one species only lands on the body of cattle, while others prefer the legs.
Response 2: For Chrysops, It was identified as Chrysops dispar, which has now been added to the manuscript. But for Tabanus, The results indicated four different species Tabanus megalops, T. rubidus, T. striatus, and T. eurytopus. They are different from each other’s by the different patterns and colors in the abdomen, which were difficult to distinguish during the experiment. With uncertainty, the results do not include the species name. However, the species information has been added into the results. (Line 163-167)
Comments 3: A further point would have been to record the light intensity in the area under investigation, as sky coverage influences the activity of the Tabanids.
Response 3: Unfortunately, light intensity was not measured. That’s an interesting point. However, we add some discussion about light intensity and feeding time of day. “Light intensity is a factor related to the biting behavior of insects. McElligott and Galloway (1991) suggest that low light intensity levels are associated with low flying activity in horse flies. The light intensity changed throughout the day; it increased during the morning, peaked at noon, and then started to decline in the afternoon, which correlates with tabanid activity.” (Line 293-297)
Comments 4: Another very interesting point would be a comparison of differently colored cattle and the effect on the attacks of tabanids.
Response 4: The cattle in the study area have the same color, (brown color). However, suggestion is importance for the further study.
Comments 5: All in all, this manuscript gives enough results to show how tabanids can be effectively kept away from cattle or how their nuisance can be reduced with a minimum of resources.
Thank you very much for your valuable comment.
Round 2
Reviewer 1 Report
Comments and Suggestions for Authors
The authors revised the manuscript based on the reviewer's comments, which clarified it further. However, I noticed some points on Figure 2. Specifically, there is a "(b)" appearing on the figure plate after previous revision. I suggest some revision or deletion of some points on Figure 2 before the paper is accepted. I have included my suggestions in the attachment.

Author Response
Comment 1: The authors revised the manuscript based on the reviewer's comments, which clarified it further. However, I noticed some points on Figure 2. Specifically, there is a "(b)" appearing on the figure plate after previous revision. I suggest some revision or deletion of some points on Figure 2 before the paper is accepted. I have included my suggestions in the attachment.
Response 1: Figure 2 was modified by removing "b", removing the headline "time of tabanid feeding on the cattle", mean±sd and other unnecessary words in the figure.